# Home-Dwelling Older Adults’ Day-to-Day Community Interactions: A Qualitative Study

**DOI:** 10.3390/geriatrics7040082

**Published:** 2022-08-12

**Authors:** Elodie Perruchoud, Armin von Gunten, Tiago Ferreira, Alcina Matos Queirós, Henk Verloo

**Affiliations:** 1Department of Nursing Sciences, School of Health Sciences, HES-SO Valais/Wallis, University of Applied Sciences and Arts Western Switzerland, Chemin de l’Agasse 5, CH-1950 Sion, Switzerland; 2Service of Old Age Psychiatry, Department of Psychiatry, Lausanne University Hospital, Route de Cery 60, CH-1008 Lausanne, Switzerland; 3Department of Health and Social Welfare, CH-1008 Lausanne, Switzerland; 4Institute of Biomedical Sciences Abel Salazar, University of Porto, 4050-313 Porto, Portugal

**Keywords:** home-dwelling older adults, behavioural disorders, local environments, workers, community network

## Abstract

Background: Many home-dwelling older adults present abnormal behaviours related to dementia or to non-dementia cognitive impairment (e.g., agitation, anxiety, apathy, etc.). Because many older adults live at home alone or are able to hide any signs of abnormal behaviours from others, the non-healthcare workers who interact with older adults on a daily basis are key actors in detecting those behaviours and orienting older adults towards appropriate support services. To the best of our knowledge, no studies to date have explored the daily interactions experienced between older adults and the various non-healthcare workers whom they regularly encounter in the community. This work aimed to identify the non-healthcare workers who are regularly in direct contact with older adults during their day-to-day activities and then develop specific training for these workers on the subject of abnormal behaviours among the elderly. Methods: This qualitative and ethnographic study asked 21 home-dwelling older adults aged 65 years old or more to answer open-ended questions. Sixteen had no self-reported cognitive impairments, and five had a probable or diagnosed slight cognitive impairment or mild-to-moderate dementia. A thematic analysis of the data was carried out. Results: The non-healthcare workers who spent the most time with older adults with and without reported cognitive impairments were those working in cafés or tea rooms and leisure or activity centres. Conclusions: In view of the significant amounts of contact between home-dwelling older adults and non-healthcare workers, it seems necessary and sensible to increase non-healthcare workers’ knowledge about abnormal behaviours, especially by offering them training. The proactive detection and identification of older adults’ abnormal behaviours by non-healthcare workers will ensure earlier care and reduce avoidable hospitalisations, institutionalisations and costs linked to geriatric healthcare.

## 1. Introduction

As the world population continues to age rapidly, more and more people will grow older and spend their final years in their own homes [1]. Many of these home-dwelling older adults can present with abnormal behaviours attributable to dementia or non-dementia cognitive impairment, such as agitation, anxiety, irritability, illusions and delusions, apathy, depression, disinhibition, and aberrant motor and obsessive compulsive behaviours, among others [2]. Several studies have underlined the numerous obstacles to identifying these abnormal behaviours, especially the facts that a large proportion of older adults live at home alone [3], are more exposed to the risk of social isolation [4] or strive to hide any signs of their abnormal behaviour from their friends and family [5,6]. In Switzerland, for example, nearly 42% of men and 87% of women aged 80 years old or more live alone, usually following the death of their partner [7]. Furthermore, as the country’s demography changes and the population ages, older adults are less likely to benefit from intergenerational support. This can be due to them having no children, or those children may be unable to care for them because they are old themselves, have their own health issues, or have professional and family commitments [7].

To avoid any increased risk of social isolation, it is essential that older adults can rely on a network of appropriate contacts within their communities. Several studies have demonstrated how social integration, especially into one’s local community, is a key element in promoting successful ageing, has a significant positive influence on older adults’ cognitive status and is an important protective factor against cognitive decline [8,9,10]. This can be evidenced as better maintenance of Mini-Mental State Examination scores and better global cognition (overall executive functioning, working memory, visuospatial abilities and processing speed) [8,9,10]. The systematic review by Kelly et al. [9] and a study in America by Holtzman et al. [8] both reported that having a suitably large social and community network promoted the maintenance of and perhaps even improvements in older adults’ cognitive status. A study in Spain by Zunzunegui et al. [10] underlined that unfavourable social relations, infrequent participation in social activities and social disengagement were some of the factors increasing the risk of cognitive decline among older adults.

No studies to date appear to have explored the interactions that older adults experience in their daily lives with the various non-healthcare workers in the community whom they regularly encounter [11]. Because home-dwelling older adults are likely to go outside their homes and interact with their neighbours in various public spaces, shops, commercial services and community activities, the local environments they frequent and the people they encounter day to day are important to know about for public health systems and policies [12,13].

An older adult’s integration in the community also represents an excellent opportunity for their interlocutors to detect abnormal behaviours when they travel and do activities outside their homes [14]. Early detection of abnormal behaviours could be a crucial factor in helping older adults remain in their own homes for as long as possible or enabling their optimal, rapid referral to formal healthcare services to assess and treat the underlying causes of their behavioural changes as quickly as possible [15].

Because detecting the early symptoms of cognitive impairment among older adult populations is recommended [16], staff working in the service sector or in local shops visited by older adults as part of their day-to-day activities could be key front-line actors in the detection of those presenting with abnormal behaviours and then in guiding them towards the appropriate social or medical support services [11].

The present study aimed to identify the non-healthcare workers with regular direct contact with older adults—people—who might notice any abnormal behaviours as older adults carry out their day-to-day activities and move around their community. The research question guiding this study was, “Who are the non-healthcare workers in the community with whom home-dwelling older adults have regular day-to-day social contact?”.

The study was performed within the framework of a call for project proposals by Health Promotion Switzerland aimed at reinforcing the quality of life of vulnerable older adults with slight cognitive impairment or mild-to-moderate dementia [17]. The CareMENS project’s aim was to promote the early detection of abnormal behaviours, probably associated with different levels of cognitive impairments, as older adults go about their daily lives. Early detection would help to maintain them in their own homes, reinforce the prevention of functional decline, encourage social interaction and improve the management of non-life-threatening emergencies in the community by orienting older adults towards the most appropriate professional medical or supportive care [17].

## 2. Materials and Methods

A qualitative-focused ethnographic research design was chosen for this study.

Focused ethnography—which originated in the field of sociological ethnography and is rooted in classic anthropological ethnography—focuses on specific elements of society [18]. Mayan (2016) defined focused ethnography as “a targeted form of ethnography led by a specific research question, conducted within a particular context or organization, among a small group of people to inform decision-making regarding a distinct problem” [19].

According to the Swiss State Secretariat for Education, Research and Innovation (SEFRI) [20], non-healthcare workers are professionals not actively involved in healthcare. Based on this definition, healthcare workers in this study included medical and nursing staff providing primary care. Therapists providing non-medical care, such as pharmacists, chiropodists and physiotherapists, were considered to be non-healthcare workers.

The following inclusion and exclusion criteria were used to develop the sample:

Inclusion criteria

Women and men aged 65 years old or more living in the community;Older adults with or without a probable or diagnosed slight cognitive impairment or mild-to-moderate dementia (cognitive decline in one or more areas such as memory, attention, executive function, learning or language);Home-dwelling older adults, living alone or not (in a house, studio or flat), in rural village or small town environments (<10,000 people), medium-sized town environments (<100,000 people) or dense urban city environments (100,000–200,000 people);Older adults living with or without homecare support services or the help of an informal/family caregiver;Older adults giving their informed consent to participate in the study;Older adults able to understand and speak French.

Exclusion criteria

Older adults who were hospitalised or institutionalised in a nursing home or a long-term care facility;Older adults with severe cognitive impairment, severe dementia or a probable or diagnosed related disease;Older adults who do not leave their homes;Older adults not domiciled in Switzerland.

An initial panel of 16 older adults without self-reported cognitive impairments was recruited in the community using a snowball sampling approach [21]. A first older adult was identified, and they passed on the contact details of a second potential participant, and so on. A second panel of five older adults was recruited, all with a slight cognitive impairment or probable or diagnosed mild-to-moderate dementia, using a purposive sampling method in coordination with the Vaud Alzheimer’s Association. All 21 individuals invited to take part in the study accepted to do so. Sampling continued until data saturation, which, in qualitative research, can be thought of as the point when no more new themes emerge as the interviews ensue [22].

Data collection took place between November 2020 and March 2021. An initial telephone call was made to participating older adults to describe the study, its objectives and data collection methods, to assure them about confidentiality and to arrange a time and place for their interview. Each participant was sent a written informed consent form by post. A good rapport between the principal investigator (EP) and the participants, as well as a suitable understanding of and commitment to the study, were indispensable to the data collection process [23].

Because of the unique situation caused by the COVID-19 pandemic, interviews with participating older adults without any diagnosed cognitive impairments were mainly carried out by telephone using the open-ended questions in the semi-structured interview guide (Appendix A). Participants with a probable or diagnosed slight cognitive impairment or mild-to-moderate dementia, however, were interviewed in their homes, using the semi-structured interview guide and in the presence of their informal family caregiver. This joint discussion allowed the informal family caregiver to complete what the participant was saying and provide additional information. This context facilitated verbal exchanges and created a trusting, safe environment.

These one-hour interviews aimed to allow the older adult to describe a typical day in their life, their normal weekday or weekend activities, the journeys they habitually take outside their home, the places they frequent, and the people with whom they have any social contact at home or in the community. Discussions maintained a friendly, informal atmosphere and allowed participants enough time to think about their answers, both elements favouring this inductive, exploratory approach. For older adults with a probable or diagnosed slight cognitive impairment or mild-to-moderate dementia, discussions in participants’ homes let the investigator immerse herself in her observations.

Two researchers (EP and HV) interpreted and analysed the data in parallel using an inductive approach without prior assumptions and predefined codes [24]. The thematic analysis of the data was carried out according to Leininger’s (2006) four phases of analysis [25]. Thematic analysis is described as “a method for identifying, analysing and reporting patterns (themes) within data” [26]. In phase one, audio-recorded data from the participant interviews were transcribed, and a first analysis began based on the research question [25]. In phase two, the data were coded, based on their recurrence, and classified using NVivo 12 QSR software. The data were then examined again in phase three to reveal any saturation of ideas or recurrent patterns of similar or different meanings and thus bring out major themes. The fourth and final phase was the interpretation and synthesis of the results of the data analysis [25]. In addition, study credibility was enhanced by ensuring that only the interviewees without a reported cognitive impairment were asked to validate the researcher’s observations and interpretations of their interview responses, and neutrality was ensured by consistently validating and clarifying the data with them [18]. The credibility of the data from participants with a probable or diagnosed slight cognitive impairment or mild-to-moderate dementia was ensured by presenting the collected data to both the participant and their relatives to ensure their joint validation of the interpretation of the data [27].

## 3. Results

### 3.1. Participants’ Characteristics

A total of 21 home-dwelling older adults aged 65 years old or more, from both towns and rural areas, participated in the study. Sixteen participants without self-reported cognitive impairments were interviewed by telephone, and five with a probable or diagnosed slight cognitive impairment or mild-to-moderate dementia were interviewed in their home in the presence of an informal family caregiver (Table 1).

### 3.2. Interview Results

The thematic analysis of the data revealed three pertinent themes: daily life; social networks; and non-life-threatening emergencies.

The theme of daily life covered the general components of older adults’ daily lives, i.e., their entourage, activities and the locations they regularly visited during a typical day. This theme allowed to establish an overall profile corresponding to the majority of community-dwelling older adults, including a description of their living environment, their activities and habits, and the places they frequented. The second theme grouped and described components of older adults’ social networks (i.e., non-healthcare workers whom they regularly met on a typical day) and their perceptions of the quality of their human exchanges. It provided a pattern of the interactions experienced by the majority of older adults living in the community, but it also allowed to explore the quality and significance of their exchanges more fully. Finally, the third theme grouped and described older adults’ experiences of non-life-threatening emergencies and their perceptions of the quality of community support in those emergencies. These three themes, taken together, provided a comprehensive overview of the habits of older adults living in the community, of the non-healthcare workers they might encounter in their daily lives and of the quality of their exchanges, as well as of the ability of non-healthcare workers to identify and respond to older adults’abnormal behaviours. The three themes also explore the quality of the network supporting older adults living in the community if they might be expected to have any disorders, such as abnormal behaviours.

#### 3.2.1. Theme 1: Daily Life

The data collected on participants’ everyday lives revealed that the vast majority of older adults (with or without cognitive impairment) had a robust social entourage, whether that involved their family or a network of friends. Most of the participating older adults (*n* = 16) were assisted by the presence of their family entourage.

“*I have a 45-year-old son and two grandchildren, aged 9 and 7. I am very happy to look after my grandchildren one afternoon a week. I have very good relationships with my family and friends, whom I see very often.”*(Participant 5)

“*I live with my daughter, who is 54 years old, since my wife died three years ago. My daughter stopped working to look after me.*”(Participant 17—moderate Alzheimer’s disease)

Most of the participating older adults (with or without cognitive impairment) had a network of close friends (*n* = 18) and a network of acquaintances made up mainly of neighbours (*n* = 12) and inhabitants from the town or village where they lived (*n* = 9).

“*Every day, I usually meet my neighbours. I’m lucky; I’ve got good neighbours. They always take the time to chat with me a little, and they always offer me their help. As I walk my dog every day, I also regularly meet people I know on the street and talk to them for a few minutes.*”(Participant 4)

“*We have a lot of friends with whom we regularly like to go to the cinema and the theatre*.”(Participant 13)

Older adults without self-reported cognitive impairments are very active people, both physically and mentally, and they like to be busy. The older adult participants took part in numerous activities, the most common of which were walking (*n* = 18), going shopping (*n* = 15), reading (*n* = 10) and looking after grandchildren or great-grandchildren (*n* = 6).

“*My wife and I look after two of our grandchildren, both 3 years old, every Thursday. We are pleased to have some youngsters at home, even if it’s a lot of work because they’ve got a lot of energy*.”(Participant 8)

“*I often do some shopping in my village grocery shop or in a supermarket. Because they don’t have all the products that I use at the village shop, I often go to the bigger shops in town*.”(Participant 10)

“*None of my days is the same. I am a very busy person. My daughters often ask me to look after my grandchildren. My life is quite fast-paced*.”(Participant 13)

The main locations frequented by older adult participants without self-reported cognitive impairments were: supermarkets in town (*n* = 12), most commonly once a week (*n* = 5); cafés and tearooms (*n* = 8); and leisure centres (group classes for fitness, swimming or tennis) (*n* = 8), mostly once or twice a week (both *n* = 4). —see Figure 1.

“*I like to do some shopping or have a coffee in the tea-room of the shopping centre not far from my home, five minutes’ walk away*.”(Participant 11)

“*Every afternoon, I do a group activity: aquafit twice a week, Tai-chi once a week, osteopathic-fitness once a week or walks*.”(Participant 15)

These participants also regularly shopped in grocery shops (*n* = 7), usually two or three times a week (both *n* = 3); collected medicine from a pharmacy (*n* = 7), about once a week (*n* = 3); and bought bread at a bakery (*n* = 7), usually every day or three times per week (both *n* = 3). —see Figure 1.

“*Every morning, on my way back from the village café, I stop to buy bread at my village bakery, and I talk to the sales ladies I know well.*”(Participant 6)

“*We usually go to the little cooperative shop in our neighbourhood*.”(Participant 14)

Finally, the older adult participants without self-reported cognitive impairments often frequented places of worship (*n* = 4), about once a week (*n* = 3); restaurants (*n* = 4), about once a week (*n* = 3); a butcher’s shop (*n* = 4), twice a week (*n* = 3); and the post office (*n* = 4), once a week (*n* = 2) or twice a month (*n* = 2). —see Figure 1.

“*I regularly see the butcher I go to, about twice a week*.”(Participant 11)

“*I often go for lunch at my usual restaurant, about twice a week, and to the post office counter every two weeks to pick up orders*.”(Participant 12)

“*We go to church every Sunday and talk with the other participants and the priest, and we have become familiar with each other. We are like a community; we all know one another*.”(Participant 14)

The locations most frequented by the panel of five older adults with a probable or diagnosed slight cognitive impairment or mild-to-moderate dementia were: cafés or tearooms (*n* = 3), mostly three times per week (*n* = 2), but also every day (*n* = 1); large supermarkets in town (*n* = 2), twice per week (*n* = 2); leisure centres such as a tennis club or a swimming pool, twice per week (*n* = 1) and every day (*n* = 1), respectively; and day centres (*n* = 2), four times per week (*n* = 1) and twice per week (*n* = 1). —see Figure 2.

“*I get up every day at 5:15 and have breakfast. At 6 o’clock, a friend picks me up in her car to meet our group of friends at the pool. We swim for 45 minutes, and then we always take time for a coffee in a tearoom right next to the pool*.”(Participant 19—moderate vascular dementia)

“*At the end of the morning, about four times a week, I attend an activity organised by the social workers in the community hall, such as gymnastics*.”(Participant 20—moderate Alzheimer’s disease)

“*I usually go for a walk three times a week with a friend, and we go for coffee in a tearoom near my house. I also go shopping with my daughter, once a week, in a supermarket*.”(Participant 21—early-stage Alzheimer’s disease)

In general, older adults without self-reported cognitive impairments, like older adults with cognitive impairments, were very well surrounded by a network of family, neighbours and friends. Cross-referencing the places frequented by the two panels of participants revealed, that the first three most frequented places were the same. The only differences between the two panels of participants were that older adults without self-reported cognitive impairments were more physically active, busier (e.g., taking care of grandchildren) and did more activities (e.g., walking, shopping, etc.).

#### 3.2.2. Theme 2: Social Networks

Older adult participants without self-reported cognitive impairments were able to identify the non-healthcare workers with whom they were regularly in contact in their day-to-day life: checkout workers at supermarkets in town (*n* = 12); café and tearoom staff and managers (*n* = 8); leisure centre staff (sports coaches, swimming instructors, dance teachers) (*n* = 8); grocery shop checkout workers (*n* = 7); pharmacists and pharmacy assistants (*n* = 7); bakery sales staff (*n* = 7); priests and pastors (*n* = 4); restaurant staff and managers (*n* = 4); butchers (*n* = 4); and post office counter workers (*n* = 4) (Figure 3).

The panel of five older adults with a probable or diagnosed slight cognitive impairment or mild-to-moderate dementia regularly came into contact with the following non-healthcare workers: café or tearoom managers and staff (*n* = 3); supermarket checkout staff in town (*n* = 2); staff in leisure centres or activity centres (*n* = 2); and facilitators or social workers in day centres (*n* = 2) (Figure 4).

As with the places frequented, the non-healthcare workers most commonly encountered by the two panels of participants were the same.

The older adult participants (with or without cognitive impairment) mainly reported having positive perceptions about their social contacts (*n* = 14).

“*I really enjoy talking to the people I meet. I think that it’s important to take the time to talk with other people; you often learn lots of things*.”(Participant 9)

“*I think I have excellent relations with the people in my neighbourhood, especially with the staff at the bakery and butcher’s shop, which are very close to my apartment. We all know each other and are happy to talk to each other*.”(Participant 18—mild memory impairment)

On the other hand, six participating older adults had a more negative perception of their social relations.

“*I don’t talk much with people I meet outside my house. I’m not much of a talker; I don’t like to bother people. We just talk about the weather*.”(Participant 1)

“*I’m a very social person. I think social relationships are what keep us alive. But I regret to see how the world has changed and become more and more stressful. People take less time to talk than they did before. All the emphasis is on money, not on human relations*.”(Participant 5) 

Finally, five older adults thought there was a real difference between the community life in a village and in a town.

“*Since I’ve lived in the city, social relations have been very different. It was different in the village. Everybody knows each other, whereas in town you’re a bit anonymous. Community life is richer in a village—I liked that. They know me well, but here in town, the checkout ladies don’t have the time*.”(Participant 2)

“*When you live in a village, you’ve known everybody for years. Besides, my children feel more reassured because I live here: they know that I’m well taken care of*.”(Participant 7)

The perceived quality of interactions differed depending on the type of non-healthcare workers who interacted with the older adult participants without cognitive impairment (Figure 5). Most of them underlined that their interactions were richer and of a moderate-to-long duration with:Bakery sales staff (*n* = 7).

“*There are three sales ladies at the bakery —all very kind.*”(Participant 9)

Café and tearoom staff and managers (*n* = 6).

“*I often go for coffee with my wife at the tearoom, where we have very good relations with the staff*.”(Participant 8)

Pharmacists and pharmacy assistants (*n* = 6).

“*With the people I meet, it’s often short discussions: we mostly talk about the weather. I sometimes have longer discussions with some people. My hairdresser and my pharmacist know me well, and we often have time for a talk.*”(Participant 9)

Grocery shop checkout staff (*n* = 5).

“*I like talking to the checkout lady at the grocery shop near my house. It’s a bit of a change of scenery, and it cheers you up*.”(Participant 7)

Leisure or activity centre staff (*n* = 4).

“*As I go to the pool every day, I talk very often with the three lifeguards who work there.*”(Participant 8)

Even though supermarket checkout staff were the non-healthcare workers with whom the older adults without self-reported cognitive impairments interacted most frequently (*n* = 12), they mostly described their exchanges with them as brief and superficial (*n* = 7).

“*The checkout ladies in my village and in the supermarkets are very kind, but I don’t stop very long to talk to them; I don’t want to disturb them. There is often a lot of people, so I just say ‘Hello’ so as not to bother them and to be quick*.”(Participant 1)

[…] “*I don’t talk much with sales staff and checkout ladies at the shopping centre. I don’t know them—there’s just a quick exchange of courtesies.*” (Participant 12)

In general, older adult participants with a probable or diagnosed slight cognitive impairment or mild-to-moderate dementia reported having richer, moderate-to-long spoken exchanges with café or tearoom staff and managers (*n* = 3); leisure or activity centre staff (*n* = 2); and day centre facilitators or social workers (*n* = 2). —see Figure 6.

“*I go to the day centre twice a week, and I go for coffee at a tearoom near my flat about three times a week. I get on well with the staff who work there; they are always cheerful.*”(Participant 17—moderate Alzheimer’s disease)

“*I know the people who work at the tennis club very well; I see them often*.”(Participant 21—early-stage Alzheimer’s disease)

A sole participant reported having only brief or superficial exchanges with one type of non-healthcare worker: supermarket checkout staff. —see Figure 6.

“*The contacts we have with the supermarket staff are superficial and brief. They don’t take the time to speak, and they’re not that attentive. We’re just passing through.*”(Participant 18—mild memory impairment)

Thus, the perceived quality of exchanges was similar for the two panels of participants. For older adults with or without cognitive impairment, supermarket staff represented the professionals with whom it was most difficult to have exchanges, especially because of the short times spent at the checkout, for example.

#### 3.2.3. Theme 3: Non-Life-Threatening Emergencies

Eight of the twelve older adult participants interviewed who had no self-reported cognitive impairment thought that the non-healthcare workers whom they were in contact with would be capable of detecting non-life-threatening emergencies in their behaviour.

“*I think people in my village would notice if there were changes in my behaviour. I think that they would notice straight away if I wasn’t up on my game*.”(Participant 7)

“*I think the people I meet in the community would be able to detect a change in my behaviour. I think that they’d say to themselves, ‘Hey! She’s not her usual self’.*”(Participant 15)

Three of the informal family caregivers of the five participants with a probable or diagnosed slight cognitive impairment or mild-to-moderate dementia reported that, to date, they had never had to deal with a non-life-threatening emergency involving their relation.

“*Until now, I have not experienced any non-life-threatening emergency related to my husband’s memory impairment.*”(Caregiver for participant 18—mild memory impairment)

“*My mother never goes very far from her sheltered accommodation; she is very careful. As soon as she can’t see the building anymore, she retraces her steps.*”(Caregiver for participant 20—moderate Alzheimer’s disease)

“*My mother has never found herself in a difficult situation. She has retained a lot of her habits. She’s never wandered off on her own, and she’s never left the cooking hobs on, for example*.”(Caregiver for participant 21—early-stage Alzheimer’s disease)

Two informal family caregivers recounted a non-life-threatening emergency involving two types of non-healthcare professionals—a bus driver and a train ticket inspector—both of whom had been able to detect and manage the older adult participant’s abnormal behaviour.

“*My father left home and took the train to his village station. Luckily, the train’s ticket inspector noticed that he was behaving unusually and that he did not have a ticket. He had also seen him get on the train and knew which station he had boarded at. The controller then reacted very well, bringing my father in to check all the other tickets with him, and then taking him back to his starting point on the way home. There, he asked some villagers about his identity and was able to contact me directly to pick him up*.”(Caregiver for participant 17—moderate Alzheimer’s disease)

“*One day, my aunt got on a bus because it stopped right next to her. Luckily, the bus terminal was just two stops further, so the driver asked her where she wanted to go and noticed that she was a little confused. Luckily, she had the presence of mind to give him her telephone and ask him to call me, which the driver did, and I was able to come and collect her*.”(Caregiver for participant 19—moderate vascular dementia)

Finally, one informal family caregiver said that she had a negative view of non-healthcare workers’ ability to act in response to an older adult’s abnormal behaviour.

“*I don’t think people in the community can notice a disorder in another person. For example, if you take my dad’s situation, he’s somebody who can answer quickly. I’m scared that if he had a problem, the people opposite him might ask him some questions, think that he is just a little strange and not take things any further*.”(Caregiver for participant 17—moderate Alzheimer’s disease)

The comments received highlighted that older adults may be confronted with non-life-threatening emergencies due to the presence of a variety of disorders. In such cases, the majority of the testimonies revealed that non-healthcare workers seemed to be able to react adequately and that they represented an effective means of support.

## 4. Discussion

This study appears to be the first ethnographic field study to explore which non-healthcare workers home-dwelling older adults aged 65 years old or more will typically have regular daily contact with in the community as well as how they maintain their mobility and where they go. The study identified the non-healthcare workers who most frequently encountered older adults without self-reported cognitive impairments in their daily lives. It also identified with whom older adults had the most meaningful exchanges: café or tearoom staff and managers; staff in leisure centres and day centres; grocery shop checkout staff; pharmacists and pharmacy assistants; and lastly, bakery sales staff. For older adults with a probable or diagnosed slight cognitive impairment or mild-to-moderate dementia, those workers were: café and tea-room staff and managers; staff in leisure centres and day centres; day centre facilitators and social workers.

In line with the studies by Evans et al. [3], Dury [4] and Monod et al. [7], the description of the study sample showed that just over half of the older adults lived at home alone. For most of the older adults interviewed, maintaining social contacts in the community was important, and they benefitted positively from it. This aspect agreed with the studies by Holtzman et al. [8], Kelly et al. [9] and Zunzunegui et al. [10], which emphasised that social integration was a key element in promoting successful ageing. Similar to the studies of Clark et al. [12], Ward et al. [13] and Brodaty et al. [14], the present study’s results also revealed that the majority of older adults regularly visit public places and meet many non-healthcare workers, providing an excellent opportunity for them to detect abnormal behaviours.

Finally, the results underlined that older adults generally do have a network of appropriate contacts within their community. Yet, this is in contrast to the commonly perceived notion that older age has a negative influence on social and community relations. However, it supports the conclusions of the study in America by Cornwell et al. [28], which reported that certain transitions linked to age, such as retirement or a bereavement, frequently led to greater engagement in the community and a richer connection to society.

## 5. Limitations

This qualitative-focused ethnographic study had some limitations. Because of the situation caused by the COVID-19 pandemic, interviews with participating older adults without diagnosed cognitive impairments were mainly carried out by telephone. Ideally, the investigator (EP) would have visited the participants in their homes to observe and interview them in their own living context, thus facilitating more valid and accurate data collection and interpretation. Telephone exchanges did not enable the observation of participants’ attitudes and non-verbal behaviours, which may have biased the interpretation of the results to some extent: the transcripts of participants’ responses may not have accurately represented the feelings and ideas expressed. Furthermore, since researchers themselves function as the main data collection tool in ethnographic research, it cannot be excluded that certain key discussion points may have been influenced by the investigator’s personal attitudes and statements. Data interpretation and analysis may have been significantly influenced by the researchers’ own representations and their cultural context. In addition, the data analysis was intended to be predominantly descriptive, which may alter the meaningful depth of the data. The sample and the study setting were restricted to a specific French-speaking area of Switzerland, which might have affected the results’ transferability. Finally, the panel of five older adults with a probable or diagnosed slight cognitive impairment or mild-to-moderate dementia was quite restricted, which may have compromised the achievement of data saturation, despite the repetition of identical themes by the two participant panels. Ensuring the validity and reliability of the results for participants with a probable or diagnosed slight cognitive impairment or mild-to-moderate dementia is a challenge [29]. To facilitate successful interviews with them, the researchers used effective communication skills and strategies. One strategy was to conduct interviews in participants’ homes, thus reducing any potential anxiety about an unfamiliar environment and minimising some distractions. The wording of the interview questions was also adapted to participants’ cognitive abilities by using simplified, familiar and understandable terminology. Moreover, participants were given sufficient time to reflect on and answer questions [27]. In addition, the researchers used several strategies to enhance the validity and reliability of the data from these participants, based on the recommendations of Lincoln and Guba (1985) [30]. To ensure the credibility and dependability of the data from participants with a probable or diagnosed slight cognitive impairment or mild-to-moderate dementia, the researchers spent sufficient time with the participants to collect enough rich data. Finally, the process of triangulating participants’ declarations, those of their relatives, and the researchers’ observations and field notes made it possible to obtain a sense of the data in its entirety [30].

## 6. Strengths

The present study had numerous strengths. The researchers used triangulation during several stages of their work [31]. During data collection, they explored the points of view of several older adult participants with different mental health profiles as well as those of several informal family caregivers. Two different interview methods were used, either telephone communication or using immersion in the participant’s environment. Because the researchers (EP and HV) coded the data and carried out their thematic analyses separately, there can be greater trust in the links revealed between the study’s data collection, analysis and interpretation. Finally, the sample’s heterogeneity allowed a variety of situations and older adult lifestyle habits in the community to be explored, whether or not they had a cognitive impairment. The rich and diverse data collected enabled the extraction of salient themes. In general, the older adult participants and their informal family caregivers were pleased to have had the opportunity to describe their daily lives and the people and places they frequented, and to share their points of view and thoughts.

## 7. Conclusions

Integration into one’s community has been shown to lead to positive health outcomes for older adults [8,9,10], and the places they frequent there represent crucial locations with opportunities for detecting abnormal behaviours related to dementia or to non-dementia-cognitive impairment [14]. Because older adults living alone in their own homes are frequently quite isolated, community life can become a determining factor in the process of detecting and orienting older adults presenting with abnormal behaviours. The present study aimed to identify the non-healthcare workers with regular direct contact with older adults—people—who might notice any abnormal behaviours as older adults carry out their day-to-day activities and move around their community. Knowing which workers are in regular contact with older adults in the community is an important indicator for public health authorities, helping them to develop health promotion and prevention interventions so that the right target groups can learn about assisting those older adults. Therefore, it seems essential that more research be carried out on home-dwelling older adults’ networks of destinations and contacts in the community in order to identify the significant non-healthcare workers with whom they interact frequently. It is also necessary to provide those non-healthcare workers with the tools to identify older adults’ abnormal behaviours. They are undoubtedly among the first to encounter older adults presenting with abnormal behaviours; they could be trained to detect and orient their fellow community members who are unsettled and unable to react to their behaviour appropriately alone [32,33]. In this project’s second phase, the results will be used to facilitate focus groups with the non-healthcare workers identified to collect their opinions and information needs concerning the disorders they might encounter among older adults and the actions they could take. Based on these results, it will then be possible to issue recommendations on the development of training courses that meet the community’s needs as closely as possible. Training could help non-healthcare workers in the community understand older adults’ abnormal behaviours and boost their ability to proactively detect them and warn professionals. It might also enable them to orient the older adults concerned towards appropriate care pathways. These training sessions would encourage the earlier identification of behavioural problems, the earlier initiation of care, the more efficient use of healthcare and home care services for older adults, as well as a reduction in avoidable hospitalisations, institutionalisations and costs linked to geriatric health. In conclusion, this study’s results should provide a great stimulus for further studies, but they also suggest further action to encourage complementary collaboration between citizens, non-healthcare workers, local authorities, elderly people and their families, healthcare services and public health decision-makers, all of whom would benefit very favourably from training about older adults’ behavioural problems.

## Figures and Tables

**Figure 1 geriatrics-07-00082-f001:**
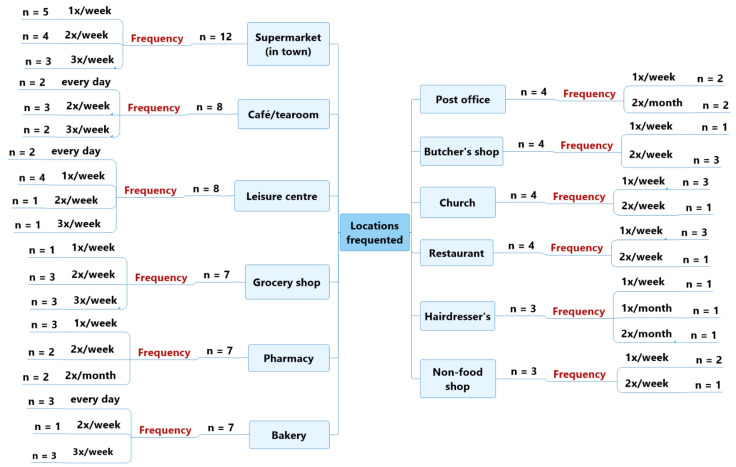
Characteristics of the locations frequented by the participants without self-reported cognitive impairments.

**Figure 2 geriatrics-07-00082-f002:**
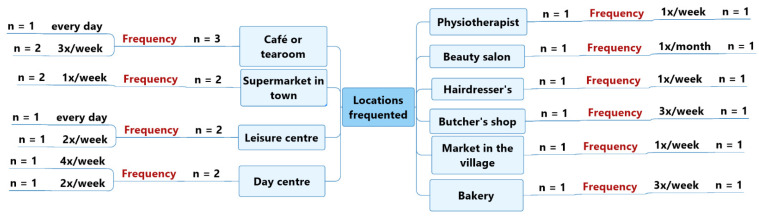
Types of locations frequented by study participants with a probable or diagnosed slight cognitive impairment or mild-to-moderate dementia.

**Figure 3 geriatrics-07-00082-f003:**
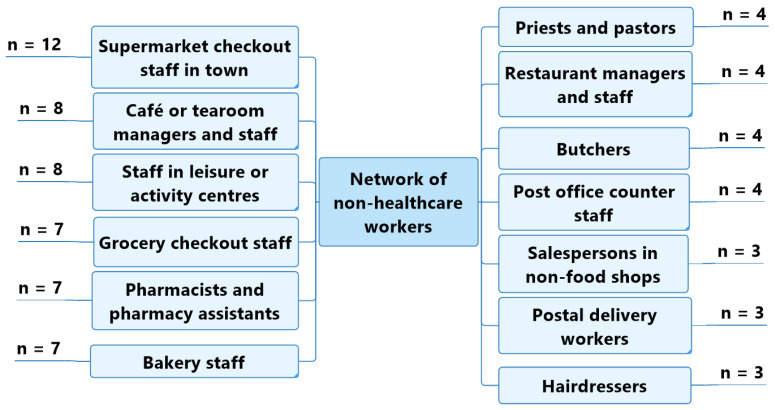
Professional characteristics of the non-healthcare workers regularly met by participants without self-reported cognitive impairment.

**Figure 4 geriatrics-07-00082-f004:**
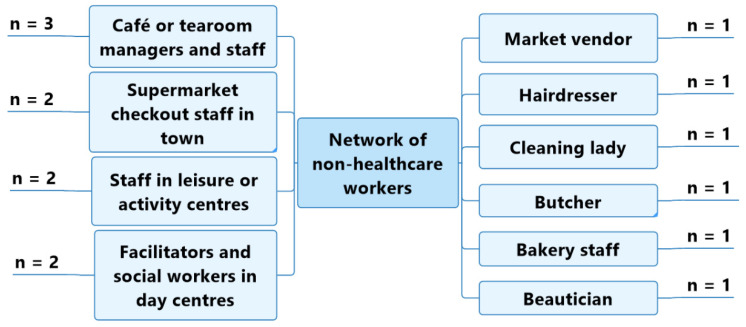
Non-healthcare workers regularly encountered by participants with a probable or diagnosed slight cognitive impairment or mild-to-moderate dementia.

**Figure 5 geriatrics-07-00082-f005:**
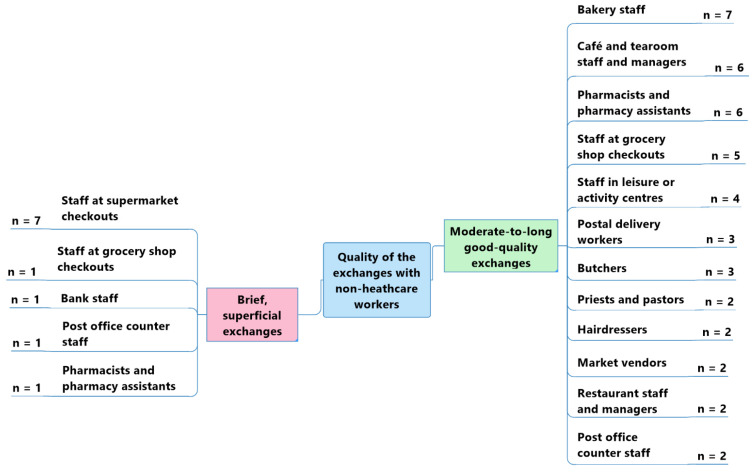
Perceived quality of the exchanges between participants without self-reported cognitive impairments and non-healthcare workers.

**Figure 6 geriatrics-07-00082-f006:**
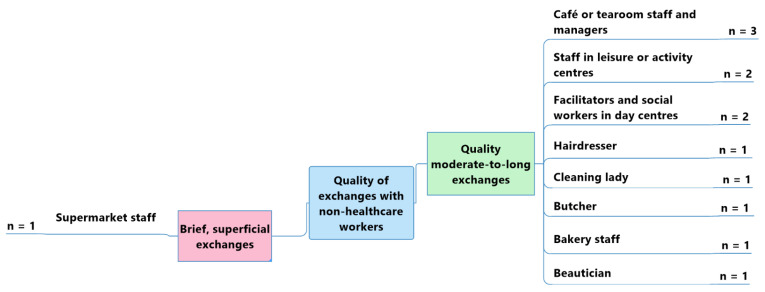
Perceived quality of the exchanges between participants with a probable or diagnosed slight cognitive impairment or mild-to-moderate dementia and non-healthcare workers.

**Table 1 geriatrics-07-00082-t001:** Participants’ characteristics.

Interview N°	Gender	Age	Diagnosis	Marital Status	Housing	Location *	Lives Alone	Family Caregiver
1	Woman	82	-	Married	House	Village	No	-
2	Woman	68	-	Widowed	Flat	Medium-sized town	Yes	-
3	Man	75	-	Married	Flat	Medium-sized town	No	-
4	Man	83	-	Widowed	Flat	Village	Yes	-
5	Woman	73	-	Married	Flat	Medium-sized town	No	-
6	Man	80	-	Married	House	Village	No	-
7	Woman	91	-	Widowed	House	Mountain village	Yes	-
8	Man	65	-	Married	Flat	Medium-sized town	No	-
9	Woman	72	-	Single	House	Medium-sized town	Yes	-
10	Man	85	-	Widowed	House	Village	Yes	-
11	Man	81	-	Widowed	Flat	Dense urban city	Yes	-
12	Man	79	-	Single	Flat	Dense urban city	No	-
13	Woman	69	-	Married	Flat	Dense urban city	No	-
14	Woman	90	-	Married	Flat	Dense urban city	No	-
15	Woman	72	-	Separated	Flat	Dense urban city	Yes	-
16	Man	72	-	Married	Flat	Dense urban city	No	-
17	Man	82	Moderate Alzheimer’s disease	Widowed	Daughter’s flat	Village	No	Daughter
18	Man	76	Mild memory impairment	Married	Flat	Village	Yes	Wife
19	Woman	86	Moderate vascular dementia	Single	Flat	Medium-sized town	Yes	Niece
20	Woman	82	Moderate Alzheimer’s disease	Widowed	Sheltered flat	Village	Yes	Daughters
21	Woman	66	Early stage Alzheimer’s disease	Widowed	House	Medium-sized town	Yes	Daughter

* Official designations of what constitutes a village or town vary by Canton. We use the following key: rural village or small town environments (<10,000 people), medium-sized town environments (<100,000 people) or dense urban city environments (100,000–200,000 people).

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
