# Peer review of "Home-Dwelling Older Adults’ Day-to-Day Community Interactions: A Qualitative Study"

_geriatrics, 2022, doi:10.3390/geriatrics7040082_

Round 1
Reviewer 1 Report
Thank you for the opportunity to review this paper. It is a worthy subject with an interesting methodological approach. I hope my comments are helpful.
Generally, the abstract is lacking detail and could be more clear and specific about both the purpose of the article, the gap in the literature it is addressing, and how it intends to do so.
· In the abstract the authors state “abnormal and hard to identify behaviour” – this type of language is a bit hard to understand especially as the first sentence and could be clearer.
· Second sentence of abstract ‘they’ referent unclear
· The purpose of the article is clear but it is unclear why this work is important.
· The ‘latter’ in line 19 of the abstract is unclear
Background is well written
Methods-
- The explanation of the methodological approach is clear but why the authors state they based their study on the COREQ criteria is not clear to me. The COREQ is a post-positivist checklist that be used to check if you have substantive aspects of qual methods included (although there have been many critiques of COREQ and of how it was created).
- Can the authors describe what saturation means in the context of this study and how it is consistent with their methodological approach?
- In line 157, I wonder if the authors truly believe that keeping a log book is a means to objectively bracket their biases? And if this is even consistent with an ethnographical approach?
Results
The themes are not well developed. I would suggest that the authors revisit their references on thematic analysis. The quotes are to be used sparingly and to demonstrate the point, rather than be the point. Thematic analysis requires that the authors actually describe the collective theme and I do not see this well developed in the results section. One suggestion would be to write out the themes in paragraph form, and then add back in quotes where they are necessary.
I would also offer that 5 themes is a lot of themes and would urge the authors to look more closely for overlap in their responses. As presently written, it is hard to discern where the overlap lies because the themes are not well developed.
Discussion
The discussion could focus more on comparing the study findings to the extant literature. Currently there is only one paragraph of discussion, one of limitation, and one of strengths. While the reflection on methodological strengths and limitations is appreciated, it would be helpful to spend more time talking about the study findings and their implications.
The limitations section should be the last paragraph before the conclusion and have a heading.
Conclusion is well written.
Author Response
Abstract
Point 1: Thank you for the opportunity to review this paper. It is a worthy subject with an interesting methodological approach. I hope my comments are helpful.
Response 1: We thank the reviewer for taking the time to review our manuscript and for providing us with relevant comments.
Point 2: Generally, the abstract is lacking detail and could be more clear and specific about both the purpose of the article, the gap in the literature it is addressing, and how it intends to do so.
Response 2: We thank the reviewer for this relevant comment. We have reworked the abstract by supporting the lack of articles on our topic in the literature and by giving more details on our study’s objective as well as on the sample (lines 14-33).
Point 3: In the abstract the authors state “abnormal and hard to identify behaviour” – this type of language is a bit hard to understand especially as the first sentence and could be clearer.
Response 3: We thank the reviewer for this constructive comment. The words "and hard to identify " have been deleted to avoid confusion, and examples of abnormal behaviour have been added (line 14).
Point 4: Second sentence of abstract ‘they’ referent unclear.
Response 4: We thank the reviewer for this relevant comment. The term "they" has been clarified (lines 16-18).
Point 5: The purpose of the article is clear but it is unclear why this work is important.
Response 5: We thank the reviewer for this constructive comment. We have reworked the abstract to put more emphasis on the study’s importance, especially in the abstract’s Conclusions section, and to show the work’s positive potential impact (lines 28-33).
Point 6: The ‘latter’ in line 19 of the abstract is unclear. Background is well written
Response 6: We thank the reviewer for this relevant comment. The term "latter" has been clarified (lines 29-30).
Methods
Point 7: The explanation of the methodological approach is clear but why the authors state they based their study on the COREQ criteria is not clear to me. The COREQ is a post-positivist checklist that be used to check if you have substantive aspects of qual methods included (although there have been many critiques of COREQ and of how it was created).
Response 7: We thank the reviewer for this constructive comment. We agree with this comment and propose to delete this wording (from lines 110-112).
Point 8: Can the authors describe what saturation means in the context of this study and how it is consistent with their methodological approach?
Response 8: We thank the reviewer for this relevant comment. An explanation of data saturation has been added (lines 135-137).
Point 9: In line 157, I wonder if the authors truly believe that keeping a log book is a means to objectively bracket their biases? And if this is even consistent with an ethnographical approach?
Response 9: We agree with this comment. Keeping a logbook is not specific to the ethnographic approach. We have, therefore, deleted this wording (lines 181-183).
Results
Point 10: The themes are not well developed. I would suggest that the authors revisit their references on thematic analysis. The quotes are to be used sparingly and to demonstrate the point, rather than be the point. Thematic analysis requires that the authors actually describe the collective theme and I do not see this well developed in the results section. One suggestion would be to write out the themes in paragraph form, and then add back in quotes where they are necessary. I would also offer that 5 themes is a lot of themes and would urge the authors to look more closely for overlap in their responses. As presently written, it is hard to discern where the overlap lies because the themes are not well developed
Response 10: We thank the reviewer for these constructive comments. The whole results section has been reworked:
- The themes have been further developed;
- Some quotes have been removed;
- A revision of the themes led us to keep three themes out of the original five;
- Figures have been reworked for clarity.
Discussion
Point 11: The discussion could focus more on comparing the study findings to the extant literature. Currently there is only one paragraph of discussion, one of limitation, and one of strengths. While the reflection on methodological strengths and limitations is appreciated, it would be helpful to spend more time talking about the study findings and their implications.
Response 11: We thank the reviewer for this relevant comment. The comparison between our results and the existing literature has been developed further (lines 580- 591), and the whole Conclusion section has been reworked by strengthening our recommendations (development of training, prevention actions).
Point 12: The limitations section should be the last paragraph before the conclusion and have a heading. Conclusion is well written.
Response 12: We agree with this comment. The discussion on limitations has been moved to a specific paragraph with the title “Limitations” (lines 593-629).

Reviewer 2 Report
Thanks for your paper with an interesting topic. But right from the start with the Title, Abstract and Introduction, I feel that your paper should become more focused, better scoped and concise. This should be improved! The whole subtheme of "care pathways" does not materialize in this paper, has not been conceptualised, is only a very small part of the results and should not be in the title; I would strongly suggest to skipp it at all. Also, the title and your research question (lines 85-87) are not aligned and not very informative on your main themes: interaction with non-healthcare workers (but "profiles" as term is a bit overdone), and older adults without or with mild/moderate cognitive impairment/dementia (terms changes all the time; this subpanel is not in the Abstract). You are not totally strict with the term "non-healthcare workers", as there pop-up some healthcare workers in your Results (physiotherapist, homecare workers, chiropodist, pharmacist).
In your Methods, it might be wise to consistently speak of two subpanels (21 older adults and 5 persons with cognitive impairments). Be more clear about how you interviewed the family caregivers (separately or combined with person from subpanel 2) and how you triangulated them (line 153-155). Did also these (combined) interviews only take 30 minutes indeed (line 132)? Hard to believe (also, the intervie guide in the appendix is quite extensive), especially in subpanel 2! I also find it hard to believe that the Results of this subpanel with only 5 persons really got saturated (you are very short on this, line 116-117). Please elaborate more, both in the Methods and the Discussion! Your inclusion criteria raise questions: why do you use the term "benefit" in the fourth and German in the last criterium (line 588 tells us that you did your research in a French-speaking area).
The Results are very desciptive and quite long and sometimes repetitive. But more severe: they only count the numbers and types of contacts but do qualify them. The structure of your Figures raise questions (why blue/green, what is difference between left/right (is it the N?; why not in figure 2?). The N in particular findings is often so low (just 1 or 2) that it raises questions about the meaningfulness of the Results. The (quotes for) theme about care path ways are not about care path ways. This all theme should be removed from the whole paper in my opinion.
As the Results only count types and numbers of contacts, over-exaggerating that with the term "profiles" (please write a limitation about this rather desciptive nature of your research), you cannot give any explications about the quality or appropriateness of the networks, in the way you did for instance in lines 569-571.
Many of the issues I raised above, should find place in you Discussion.
Perhaps you could consider a new part with recommendations for practice and further research.
Author Response
Point 1: Thanks for your paper with an interesting topic. But right from the start with the Title, Abstract and Introduction, I feel that your paper should become more focused, better scoped and concise. This should be improved!
Response 1: We thank the reviewer for taking the time to review our manuscript and for providing us with relevant comments. We have reworked the abstract with supporting information because of the lack of articles on our topic in the literature and have given more details on our study’s objectives and the sample (lines 14-34). The Introduction section has also been reworked (see response 3).
Point 2: The whole subtheme of "care pathways" does not materialize in this paper, has not been conceptualised, is only a very small part of the results and should not be in the title; I would strongly suggest to skipp it at all.
Response 2: We agree with this comment and have removed the subtheme of “care pathways” from the title.
Point 3: Also, the title and your research question (lines 85-87) are not aligned and not very informative on your main themes: interaction with non-healthcare workers (but "profiles" as term is a bit overdone), and older adults without or with mild/moderate cognitive impairment/dementia (terms changes all the time; this subpanel is not in the Abstract). You are not totally strict with the term "non-healthcare workers", as there pop-up some healthcare workers in your Results (physiotherapist, homecare workers, chiropodist, pharmacist).
Response 3: We thank the reviewer for these constructive comments. We have reworked these different points by:
- deleting the paragraph on care pathways, which is not relevant to our study’s purpose (lines 66–69)
- keeping the study to a single objective for clarity and conciseness (lines 90-99)
- deleting the word "profiles" throughout the text
- naming the two sample panels in the abstract (lines 23-25)
- clarifying the definition of non-healthcare workers in the Methods’ section (lines 119-124)
- deleting the term “homecare workers” from Figure 3 as it is confusing
Point 4: In your Methods, it might be wise to consistently speak of two subpanels (21 older adults and 5 persons with cognitive impairments). Be more clear about how you interviewed the family caregivers (separately or combined with person from subpanel 2) and how you triangulated them (line 153-155). Did also these (combined) interviews only take 30 minutes indeed (line 132)? Hard to believe (also, the interview guide in the appendix is quite extensive), especially in subpanel 2!
Response 4: We thank the reviewer for these relevant comments, which we agree with. We have reinforced the distinction between the two participant panels in the abstract’s Methods section (lines 129-162). In addition, we have provided more details about family caregivers’ involvement with participants with cognitive impairments and about the triangulation (lines 151-153; lines 177-180). Finally, it is true that the initial intention was to carry out 30-minute interviews, but these often lasted an hour. We have changed the time frame to be more realistic (line 154).
Point 5: I also find it hard to believe that the Results of this subpanel with only 5 persons really got saturated (you are very short on this, line 116-117). Please elaborate more, both in the Methods and the Discussion! Your inclusion criteria raise questions: why do you use the term "benefit" in the fourth and German in the last criterium (line 588 tells us that you did your research in a French-speaking area).
Response 5: We thank the reviewer for this constructive comment. Indeed, reaching data saturation is difficult with a limited panel of 5 older adults, as we state in the study limitations section (lines 610-613). In addition, we have added an explanation of data saturation (lines 135-137). Concerning the inclusion criteria, the word “German” (line 128) was removed as indeed you noted that all the participants were French-speaking. Finally, the word “benefit” was replaced to improve understanding.
Point 6: The Results are very desciptive and quite long and sometimes repetitive. But more severe: they only count the numbers and types of contacts but do qualify them. The structure of your Figures raise questions (why blue/green, what is difference between left/right (is it the N?; why not in figure 2?). The N in particular findings is often so low (just 1 or 2) that it raises questions about the meaningfulness of the Results. The (quotes for) theme about care path ways are not about care path ways. This all theme should be removed from the whole paper in my opinion. As the Results only count types and numbers of contacts, over-exaggerating that with the term "profiles" (please write a limitation about this rather desciptive nature of your research), you cannot give any explications about the quality or appropriateness of the networks, in the way you did for instance in lines 569-571. Many of the issues I raised above, should find place in you Discussion.
Response 6: We thank the reviewer for these relevant comments. The whole Results section has been reworked:
- The themes have been more fully developed
- Some quotes have been removed
- The revision of the themes has resulted in only three of the five themes being retained and the theme on care pathways has been removed
- Figures have been reworked for clarity
- The restricted number of subjects has been noted as being a limitation (lines 608-613)
- The descriptive nature of the analysis has been indicated as a limitation (lines 607-608)
- The word “profiles” has been removed throughout the text
Point 7: Perhaps you could consider a new part with recommendations for practice and further research
Response 7: We agree with this comment. The whole conclusion section (lines 647-679) has been reworked by strengthening the recommendations concerning research and practice (development of training, prevention actions, …).

Reviewer 3 Report
I appreciated your work on this important topic. I liked that you used a mixed method approach because in your research, I believe the quotes are very meaningful! A few thoughts for you to consider: (1) the font used in the figures is small - enlarging or changing the font type would be good for the reader; (2) for the tables, there is too much white space - you really need to cluster information together so the content is easier to follow; (3) I would love for the conclusion to have more meaning and depth - for example, how would you go about educating/training individuals in the community who frequently interact with older adults? I think also you could offer some innovative/visionary ideas about how utilizing community individuals would make the health of both older adults and the public health of the community more meaningful and most likely decrease health care costs. It is good for us to have this information - how can those of us caring for older adults use your data to transform how older adults living in the community can benefit from your research findings? How can we make your data have an impact on the older adults? Think outside of the box based on your findings. I hope you will consider these suggestions. Best Wishes in Your Future Research!
Author Response
Point 1: I appreciated your work on this important topic. I liked that you used a mixed method approach because in your research, I believe the quotes are very meaningful!
Response 1: We thank the reviewer for taking the time to review our manuscript and for providing us with relevant comments.
Point 2: A few thoughts for you to consider: (1) the font used in the figures is small - enlarging or changing the font type would be good for the reader; (2) for the tables, there is too much white space - you really need to cluster information together so the content is easier to follow;
Response 2: We agree with this comment. The figures and tables have been reworked for clarity.
Point 3: I would love for the conclusion to have more meaning and depth - for example, how would you go about educating/training individuals in the community who frequently interact with older adults? I think also you could offer some innovative/visionary ideas about how utilizing community individuals would make the health of both older adults and the public health of the community more meaningful and most likely decrease health care costs. It is good for us to have this information - how can those of us caring for older adults use your data to transform how older adults living in the community can benefit from your research findings? How can we make your data have an impact on the older adults? Think outside of the box based on your findings. I hope you will consider these suggestions. Best Wishes in Your Future Research!
Response 3: We thank the reviewer for this constructive comment. The whole Conclusion section has been reworked, including strengthening the recommendations concerning research and practice (development of training, prevention actions, …).

Round 2
Reviewer 1 Report
The manuscript is much improved but still has a ways to go. I hope my comments are useful.
1. In table 1- consider defining the town sizes (sm, medium, large) in a key at the bottom and then write this under the table. It is also a bit confusing that you use the term 'town' to refer to municipalities of all sizes.
2. In section 3.2 please provide an overview and not just a list of the themes. Together what do these themes mean? It is helpful to remind the reader.
3. The reduction of quotes is much improved. The quotes still feel like they are not well introduced. The authors could work on this as current;y they do feel like they are just dropped in. You need to tell the reader who said what and why. i.e. page 14 the three quotes from people with mild CI: how is this qualitative data useful? What does it add?
4. I like a short discussion section but one paragraph is insufficient. Any manuscript guidelines will tell you that the discussion should be three paragraphs at a minimum.
Author Response
Point 1: The manuscript is much improved but still has a ways to go. I hope my comments are useful.
- In table 1- consider defining the town sizes (sm, medium, large) in a key at the bottom and then write this under the table. It is also a bit confusing that you use the term 'town' to refer to municipalities of all sizes.
Response 1: We thank the reviewer for taking the time to review our manuscript once more and providing us with relevant comments. We have added a town-size key to Table 1 consistent with our inclusion criteria.
Point 2: In section 3.2 please provide an overview and not just a list of the themes. Together what do these themes mean? It is helpful to remind the reader.
Response 2: We thank the reviewer for this relevant comment. Section 3.2 has been reworked with a definition of each theme, but also with an explanation of the relationship between the themes and their purpose (lines 206-223).
Point 3: The reduction of quotes is much improved. The quotes still feel like they are not well introduced. The authors could work on this as current;y they do feel like they are just dropped in. You need to tell the reader who said what and why. i.e. page 14 the three quotes from people with mild CI: how is this qualitative data useful? What does it add?
Response 3: We thank the reviewer for this constructive comment. Throughout the text, we have specified participants’ profiles more precisely and the characteristics of the people quoted (line 231; line 241; line 243; line 255; line 319; line 322; line 327; line 368; line 376; line 445; line 448; line 455; line 489; line 492; line 497; line 509; line 515; line 524)
We have also clarified each theme in relation to participants with and without disorders (lines 331-337; lines 365-366; lines 462- 465; lines 527-530)
Point 4: I like a short discussion section but one paragraph is insufficient. Any manuscript guidelines will tell you that the discussion should be three paragraphs at a minimum.
Response 4: We thank the reviewer for this relevant comment. We have further developed and modified the Discussion section into three paragraphs (lines 532-559)
Reviewer 2 Report
The authors carefully reworked their paper and addressed comments and suggestions from the first review round adequadtely. Thanks for your efforts en reply letter.
Author Response
The authors carefully reworked their paper and addressed comments and suggestions from the first review round adequately. Thanks for your efforts in reply letter.
Response: We thank the reviewer for taking the time to review our manuscript once more and for this very positive comment.
Reviewer 3 Report
Thank you for your revisions. I just have a few recommendations!
You need to use 3rd person throughout the document - our, we, her, herself are 1st person words; the team, the authors, the research team, etc. would be 3rd person - please make those changes throughout your document
The inclusion/exclusion table has too much white space between ongoing sentence; need to reduce this so the table is readable
Cognitive impairment needs to be defined by you - diagnosis of? Memory loss? Memory issues? How are you determining the participant has CI?
For table 1 I would change heading from "sex" to "gender"
The font size in your figures is still a little small; see how much larger you can make for ease of reading
With no - without
It would be wonderful if you could show your passion about this topic in the last paragraph! What are you hoping to spark in everyone who reads your article? This is a great idea - what can others do in their community to replicate what your team has found? Think outside the box; predict a better future for older adults because of this finding!
I have included my handwritten comments on your document which should make it easy to make the revisions.
Thank you & best wishes!

Author Response
Point 1: Thank you for your revisions. I just have a few recommendations!
You need to use 3rd person throughout the document - our, we, her, herself are 1st person words; the team, the authors, the research team, etc. would be 3rd person - please make those changes throughout your document
Response 1: We thank the reviewer for taking the time to review our manuscript once more and for providing us with relevant comments. We have made the changes throughout the document.
Point 2: The inclusion/exclusion table has too much white space between ongoing sentence; need to reduce this so the table is readable
Response 2: We agree with this comment. To avoid white spaces, the table has been removed and the inclusion and exclusion criteria have been stated in sentences (lines 115 - 136).
Point 3: Cognitive impairment needs to be defined by you - diagnosis of? Memory loss? Memory issues? How are you determining the participant has CI?
Response 3: We thank the reviewer for this constructive comment. A more detailed explanation of cognitive impairment has been added (lines 118-119).
Point 4: For table 1 I would change heading from "sex" to "gender"
Response 4: We have made the proposed change (line 198).
Point 5: The font size in your figures is still a little small; see how much larger you can make for ease of reading.
Response 5: We thank the reviewer for this constructive comment. The font size of the figures has been increased to its maximum.
Point 6: With no – without
Response 6: We have made changes throughout the document
Point 7: It would be wonderful if you could show your passion about this topic in the last paragraph! What are you hoping to spark in everyone who reads your article? This is a great idea - what can others do in their community to replicate what your team has found? Think outside the box; predict a better future for older adults because of this finding!
Response 7: We thank the reviewer for this relevant comment. We have added a final paragraph at the end of the Conclusion to further highlight the positive impact of our study topic (lines 643-647).